# The RAL Small G Proteins Are Clinically Relevant Targets in Triple Negative Breast Cancer

**DOI:** 10.3390/cancers16173043

**Published:** 2024-08-31

**Authors:** David Han, Jonathan M. Spehar, Dillon S. Richardson, Sumudu Leelananda, Prathik Chakravarthy, Samantha Grecco, Jesse Reardon, Daniel G. Stover, Chad Bennett, Gina M. Sizemore, Zaibo Li, Steffen Lindert, Steven T. Sizemore

**Affiliations:** 1Department of Radiation Oncology, Arthur G. James Comprehensive Cancer Center, The Ohio State University, Columbus, OH 43210, USA; david.han@osumc.edu (D.H.); jonathan.spehar@osumc.edu (J.M.S.); dillon.richardson@osumc.edu (D.S.R.); prathik.chakravarthy@osumc.edu (P.C.); samantha.grecco@osumc.edu (S.G.); jesse.reardon@osumc.edu (J.R.); gina.sizemore@osumc.edu (G.M.S.); 2Anagenex, 20 Maguire Rd. Suite 302, Lexington, MA 02421, USA; sumudu@anagenex.com; 3Department of Internal Medicine, Arthur G. James Comprehensive Cancer Center, The Ohio State University, Columbus, OH 43210, USA; daniel.stover@osumc.edu; 4Drug Development Institute, Arthur G. James Comprehensive Cancer Center, The Ohio State University, Columbus, OH 43210, USA; chad.bennett@osumc.edu; 5Department of Pathology, Wexner Medical Center, The Ohio State University, Columbus, OH 43210, USA; zaibo.li@osumc.edu; 6Department of Chemistry and Biochemistry, The Ohio State University, Columbus, OH 43210, USA; lindert.1@osu.edu

**Keywords:** breast cancer, TNBC, HER2+, chemotherapy, RALA, RALB, RAS, BQU57, RBC8, OSURALi

## Abstract

**Simple Summary:**

Here, we demonstrate an essential role for the RAS downstream effectors RALA and RALB in TNBC and describe a promising novel small molecule RAL inhibitor. We report that TNBC, but not HER2+BC, cell lines are dependent upon RAL expression for growth in vitro and in vivo. Surprisingly, we found little correlation between RAL-dependency and the cytotoxicity of commercially available RAL inhibitors, suggesting that these inhibitors kill through effects other than RAL inhibition. Finally, we identified a novel small molecule RAL inhibitor, OSURALi, which is more toxic to RAL-dependent TNBC cell lines than RAL-independent HER2+BC or normal cell lines. Our results highlight the RALs as key molecular targets in TNBC and introduce a novel RAL inhibitor.

**Abstract:**

Breast cancer (BC) is the most frequent cancer and second-leading cause of cancer deaths in women in the United States. While RAS mutations are infrequent in BC, triple-negative (TN) and HER2-positive (HER2+) BC both exhibit increased RAS activity. Here, we tested the RAS effectors RALA and RALB, which are overexpressed in BC, as tractable molecular targets in these subtypes. While analysis of the breast cancer patient sample data suggests that the RALs are associated with poor outcome in both TNBC and HER2+ BC, our in vivo and in vitro experimental findings revealed the RALs to be essential in only the TNBC cell lines. While testing the response of the BC cell lines to the RAL inhibitors RBC8 and BQU57, we observed no correlation between drug efficacy and cell line dependency on RAL expression for survival, suggesting that these compounds kill via off-target effects. Finally, we report the discovery of a new small molecule inhibitor, OSURALi, which exhibits strong RAL binding, effectively inhibits RAL activation, and is significantly more toxic to RAL-dependent TNBC cells than RAL-independent HER2+ and normal cell lines. These results support the RALs as viable molecular targets in TNBC and the further investigation of OSURALi as a therapeutic agent.

## 1. Introduction

In the United States, over 10% of all women will be diagnosed with breast cancer (BC) over the course of their lifetime, and despite continuing diagnostic and therapeutic advances, BC remains the second leading cause of cancer death for women [1]. Subtypes of BC are broadly categorized according to the presence of several biomarkers that determine the course of therapy and are predictive of outcome. The three clinical subtypes of BC include hormone receptor positive (HR+), human epidermal growth factor receptor 2 positive (HER2+), and triple negative breast cancer (TNBC), which does not strongly express estrogen receptors, progesterone receptors, or HER2. HR+ BC is the most diagnosed subtype and generally has the best prognosis with a total five-year survival rate of 95% and a 100% five-year survival rate for localized disease [2]. The overall five-year survival drops to 91% for HER2+, that is, also HR+, 86% for HER2+/HR-disease, and down to only 78% for TNBC. Prognosis for women with metastatic TNBC and HER2+ disease is even more dismal with five-year survival rates of only 12.8% for TNBC and 40–45% for HER2+ disease [2]. Systemic treatment for TNBC has long relied upon a rigorous course of cytotoxic chemotherapy, although more recent novel therapies such as antibody–drug conjugates (ADCs) and immune checkpoint inhibitor (ICI) monotherapy have emerged to complement traditional treatment strategies [3]. Relative to TNBC, HER2+ BC is more amenable to treatment due to the availability of effective HER2-targeting therapies. Since its 1998 FDA approval, trastuzumab, a humanized anti-HER2 monoclonal antibody, has been used as an effective therapy for HER2+ BC alongside other drugs directly inhibiting HER2 activity [4]. However, the development of resistance to trastuzumab and metastatic recurrence remain significant hurdles to the further improvement in outcomes for this BC subtype. To improve the long-term outcomes for both TNBC and HER2+ BC, new and better treatment options are needed.

RAS is an important oncogene implicated in a variety of cancers; the members of the RAS family including KRAS, NRAS, and HRAS maintain vital roles in cell cycle regulation and differentiation, and activating RAS mutations supports malignant phenotypes [5]. While RAS mutations are rare in BC, NRAS is overexpressed in basal-like BC [6] and KRAS is activated by EGFR and HER2, which are overexpressed in TNBC and HER2+ BC, respectively [7,8]. Increased RAS pathway signaling is correlated with worse prognosis in breast tumors and has been found to occur frequently in aggressive BC across subtypes [7,9]

RAS downstream effectors are important targets in cancer and their inhibition has the potential to allow for therapeutic precision against the disrupted signaling pathways present in cancer cells. The two highly homologous small GTPases, RAS like Proto-oncogene A (RALA) and B (RALB), activated downstream of RAS, have been implicated in tumor growth and metastasis in a wide variety of cancer types including colorectal cancer, lung cancer, melanoma, and BC [10]. The RALs are activated by RAL guanine nucleotide exchange factors (RALGEFs), which facilitate the binding of GTP to the RALs. Several RALGEFs including RALGDS [11], RGL1 [12], RGL2 [13], and RGL3 [14] are activated by RAS. GTP-bound RALs can be inactivated by RAL GTPase-activating protein (RALGAP) complexes, which encourage the hydrolysis of GTP to GDP. Loss of RALGAPs have been found to drive aggressive phenotypes in a pancreatic ductal adenocarcinoma model [15]. Downstream RAL effectors include RALBP1, which regulates mitochondrial fission [16] and endocytosis [17,18], PLD1, which modulates endosome trafficking [19], and SEC5 and EXO84, which are members of the exocyst complex [20].

RALA and RALB adopt a plethora of roles in cancer, from supporting tumor growth to mediating invasion and metastasis [10]. In certain cancers, such as pancreatic cancer [21,22], melanoma [23], and lung cancer [24,25], both RALA and RALB have been found to promote tumorigenicity. For other cancers, it appears that a single RAL may predominate to support aggressive phenotypes, as in the renal cancer [26] and prostate cancer cell lines [27]. However, in colorectal cancer, the RALs appear to have opposite roles, with stable shRNA knockdown of RALA suppressing and RALB knockdown enhancing anchorage-independent growth in vitro [28]. Opposing functions between the paralogs were also observed in the UMUC-3 bladder cancer cell line, in which constitutively active RALA inhibited cell migration while the reverse occurred in response to constitutively active RALB [29].

Divergence in the roles of RALA and RALB has also been observed in BC. Previous in vitro and in vivo experiments by our group using MDA-MB-231 TNBC cell lines modified by CRISPR to lack RALA or RALB have demonstrated that RALA drives tumorigenicity while RALB opposes growth and invasion [30]. Our analysis of BC patient tumor gene expression using the large METABRIC cohort [31] found that elevated RALA expression was associated with poor outcomes while elevated RALB expression was associated with a more favorable prognosis [30]. Other work using stable shRNA-mediated RAL knockdown in 4T1 mouse BC cells identified RALB as a driver of primary tumor growth, while both RALs were reported to support lung metastasis in this model [19]. The variety and occasional discordance in the apparent roles of RALA and RALB both within and between cancer types suggest that diverse roles are commonly assumed by the two RALs in cancer. As such, a more thorough investigation of the RALs across BC subtypes is needed to better understand their potentially conflicting roles in cancer.

In response to the difficulties encountered in directly inhibiting RAS, targeting of RALA and RALB has emerged as a tractable alternative within the RAS pathway, inspiring the development of small molecule inhibitors (SMIs) [32] and stapled peptides [33,34] targeting the RALs. Previous efforts to target RALA and RALB with small molecule inhibitors identified the compounds RBC6, RBC8, and RBC10 to be capable of reducing RALA activation by binding an allosteric site on GDP-bound RAL [32]. BQU57, a derivative of RBC8, disrupted the anchorage-independent growth of human lung cancer cells in vitro and substantially inhibited RALA and RALB activity and tumor growth in vivo [32]. However, the currently available RAL inhibitors RBC8 and BQU57 are potentially limited by undesirable off-target effects and poor physiochemical properties [35,36]. Thus, the discovery of improved RAL-targeting compounds is necessary to allow RAL inhibition to be a clinically appropriate strategy.

Following our observation that survivorship across BC, and especially in TNBC, is linked to the RALs, we herein explore the requirements of RALA and RALB in supporting cancer cell viability and tumor growth across a panel of BC cell lines with special focus granted to TNBC and HER2+ BC. In orthotopic tumor models, we found that the RAL dependency varied substantially between cell lines and BC subtypes. Stable knockdown of either RALA or RALB decreased tumor growth in the TNBC MDA-MB-468 line but had no impact upon tumor growth in the HER2+ SKBR3 cell line. Interestingly, decreased tumor growth in RAL-depleted MDA-MB-468 lines is driven by changes in the tumor microenvironment associated with RAL-depleted tumors rather than by changes in tumor cell viability. Using siRNAs to transiently deplete the RALs in a panel of BC cell lines, we found that the TNBC cell lines were dependent upon RAL expression for viability while the HER2+ lines were not RAL-dependent. We also examined the efficacy of two commercially available RAL inhibitors, BQU57 and RBC8, in the TNBC and HER2+ BC cell lines. Surprisingly, we found little correlation between sensitivity to these inhibitors and a cell line’s dependence upon RAL expression for viability, suggesting that these inhibitors kill cancer cells primarily via off-target effects. Finally, we report our discovery of a novel RAL inhibitor, OSURALi, which exhibits strong affinity to RALA and inhibits both RALA and RALB GTP-binding. Importantly, OSURALi is significantly more cytotoxic to RAL-dependent TNBC cell lines than RAL-independent HER2+ cell lines or normal cells, suggesting that it decreases cancer cell viability largely through on-target effects. Together, our findings demonstrate the particular importance of RALA and RALB in the context of TNBC and introduce a new RAL inhibitor that may serve as the basis for targeted therapies to treat this aggressive BC subtype.

## 2. Materials and Methods

### 2.1. Cell Culture

MDA-MB-231, BT-474, BT-549, MDA-MB-468, MDA-MB-453, and SKBR3 human breast carcinoma cell lines, HMEC-1 human foreskin endothelial-like cells, and HMECS (Human Mammary Epithelial Cells) were obtained from ATCC. MDA-MB-231, BT-474, and BT-549 cells were cultured in RPMI-1640, MDA-MB-468 and MDA-MB-453 cells in DMEM, and SKBR3 cells in McCoy’s media. HMEC-1 cells were cultured in MCDB131 media supplemented with 10 ng/mL EGF and 1 μg/mL hydrocortisone. HMECS were cultured in Mammary Epithelial Cell Basal Media (ATCC PCS-600-030) with the Mammary Epithelial Cell Growth Kit (ATCC PCS-600-040). All cell lines except HMECS were supplemented with 10% fetal bovine serum (FBS), 2% pen strep, and 1% L-glutamine and incubated at 37 °C with 5% CO_2_.

### 2.2. siRNA Knockdown

Transient knockdown of RALA and RALB was performed in MDA-MB-231, MDA-MB-468, MDA-MB-453, and BT-549 cell lines using 50 pmol of siRNA. In the BT-474 and SKBR3 cell lines, 100 pmol of siRNA (ON-TARGETplus, Dharmacon, Lafayette, CO, USA) was used. The following sequences were used to target human RALA: 5′-GGACUACGCUGCAAUUAGA-3′ (siRALA-1), 5′-CAAAUAAGCCCAAGGGUCA-3′ (siRALA-2), and 5′-GCAGACAGCUAUCGGAAGA-3′ (siRALA-3). The sequences used to target human RALB were: 5′-UCACAGAACAUGAAUCCUU-3′ (siRALB-1), 5′-GAAACAAGUCUGACCUAGA-3′ (siRALB-2), and 5′-GAAAGAUGUUGCUUACUAU-3′ (siRALB-3). Transfection of cells was performed using Lipofectamine RNAiMax (Thermo Fisher Scientific, Waltham, MA, USA) in 6-well plates. Media were changed 24 h post-transfection and the cells were incubated for an additional 24 h. Cells were then detached with trypsin and replated for viability assays.

### 2.3. Lentivirus Transduction

MDA-MB-468 parental cells were transduced with the Origene lentiviruses shCTRL (TR30021V), shRALA (TL309957VC, 5′-CTGGTTGGTAACAAATCAGATTTAGAAGA-3′), and shRALB (TL309956VD, 5′-GAACAGATTCTCCGTGTGAAGGCTGAAGA-3′). SKBR3 parental cells were grown as described and transduced with lentivirus purchased from Vectorbuilder (Chicago, IL, USA): shCTRL (5′-CCTAAGGTTAAGTCGCCCTCG-3′), shRALA (5′-CTGGTTGGTAACAAATCAGATTTAGAAGA-3′), and shRALB (5′-GGACAAGGTGTTCTTTGACCTAATGAGAG-3′). Cells were incubated with the virus for 24 h, then fresh media were added. After 3–4 days, transduced cells were selected with 5 µg/mL puromycin for 14 days. Validation of knockdown was performed via Western blot prior to the experiments.

### 2.4. Animal Procedures

Animal care and experiments were performed in compliance with University Laboratory Animal Resources (ULAR) regulations under the OSU Institutional Animal Care and Use Committee (IACUC)-approved protocol 2007A0120-R5. Female NOD scid γ (NSG) mice were acquired from the NSG mouse colony maintained by the Target Validation Shared Resource (TVSR) at Ohio State University; breeders (strain #005557) for the colony were received from the Jackson laboratory. For orthotopic fat pad injections, 5 × 10^6^ MDA-MB-468 or 3 × 10^6^ SKBR3 cells were injected into 8–12 week old NSG mice. For the SKBR3 orthotopic tumor experiments, cells were injected in a 1:1 mixture of PBS:Matrigel. Once palpable tumors were detected, two-dimensional caliper measurements of tumor size were taken twice weekly. Tumor volume was calculated as:½ (length × width^2^),
with the shorter dimension designated as the width.

Cancer cell viability was tested before and after in vivo injections on a Countess III Automated Cell Counter (Thermo Fisher Scientific) using 4% trypan blue to stain the living cells. For the MDA-MB-468 tumors, mice were sacrificed once the early removal criteria (ERC) were met and the tumor tissue was collected. For the SKBR3 tumors, all mice were sacrificed 100 days after tumor cell injection. Three independent tumor growth experiments were conducted using the MDA-MB-468 model with a total of 20 mice per group combined. The SKBR3 tumor growth data were from a single experiment of 9–10 mice per group.

### 2.5. In Vitro Viability Assays

The Cell Proliferation Kit I (Millipore-Sigma, Burlington, MA, USA) MTT assay was used to determine 2D viability in vitro. Cells were plated at 2000 cells per well in clear flat-bottom 96-well plates (Corning, Corning, NY, USA) with four technical replicates per group. For the drug EC50 assays, DMSO, RBC8, BQU57, or OSURALi solutions were added to cell suspension aliquots immediately prior to plating at the indicated concentrations in a 0.5% DMSO vehicle. For all MTT experiments, absorbance readings at 560 nm were recorded after 72 h.

For growth in low adhesion (GILA) assays measuring 3D viability, 2000 cells per well were seeded into 96-well clear round-bottom ultra-low attachment plates (Costar) in 100 μL of complete media with four technical replicates per group. Drug EC50 assays were plated as in the MTT protocol. After 5 days, 100 μL of the CellTiter-Glo 3D Viability Assay reagent (Promega, Fitchburg, WI, USA) was added to each well. After rocking for 25 min, 150 μL of lysate from each well was moved to a white polystyrene 96-well plate (Corning). Luminescence readings were recorded at 1 s using a Glomax Discover microplate reader (Promega).

### 2.6. In Vitro Migration and Invasion Assays

Transwell inserts (6.5 mm) with 8.0 µm pore polycarbonate membranes (Corning) and BioCoat Matrigel Invasion Chamber inserts with 8.0 µm PET membranes (Corning) were used to study cell migration and invasion, respectively. Before plating, cells were serum-starved overnight. For the migration experiments, cells were seeded at a concentration of 50,000 cells in 100 μL of serum free media in the interior of the migration chamber, and 600 μL of serum supplemented media was added to the bottom of the wells. For the invasion experiments, cells were seeded at 50,000 cells in 500 μL of serum-free media in the interior of the insert, and 750 μL of serum supplemented media was added to the bottom of the wells. MDA-MD-468 cells were allowed to migrate at 37 °C with 5% CO_2_ for 6 h, while SKBR3 cells migrated for 24 h. All invasion incubations lasted 24 h. After the allotted time, cells that did not migrate were removed from the upper portions of the membranes using cotton swabs. The cells on the bottom side of the inserts were then fixed in Diff-Quick fixative and stained in Diff-Quick Stain 1 and Diff-Quick Stain 2 (Polyscience, Niles, IL, USA) in succession for 5 min each before being rinsed in water. After drying, chambers were imaged and cells were manually counted for quantification.

### 2.7. RNA Extraction

RNA isolation was performed by adding TRIzol (1 mL per 0.1 g of cells) to a confluent plate of cells and incubating for 5 min at room temperature. Chloroform (0.2 mL per 1 mL of TRIzol) was added and the sample was incubated for 2–3 min at room temperature. The samples were centrifuged (12,000× *g*, 15 min, 4 °C), and the RNA from the aqueous phase was collected. The precipitation of RNA was performed with the addition of isopropanol (0.5 mL per 1 mL of TRIzol) and incubation at room temperature for 10 min. The samples were then centrifuged (12,000× *g*, 10 min, 4 °C), and the supernatant was decanted. The cell pellet was resuspended in 75% ethanol (1 mL/1 mL of TRIzol) and centrifuged (7500× *g*, 10 min, 4 °C). The supernatant was discarded, and the cell pellet was dried. RNA pellets were resuspended and treated with DNase.

### 2.8. Quantitative RT-PCR

Isolated RNA was converted to cDNA using a Maxima First Strand cDNA Synthesis Kit (Thermo Fischer Scientific, Catalog # K1641). The Taqman assay (Applied Biosystem, Waltham, MA, USA) was used to perform RT-PCR to quantify RALGDS (Thermo Fischer Scientific, Catalog # Hs00325141_m1), RALGPS1 (Thermo Fischer Scientific, Catalog # Hs01115437_m1), RALGPS2 (Thermo Fischer Scientific, Catalog # Hs00404163_m1), RCC2 (Thermo Fischer Scientific, Catalog # Hs00603046), RGL1 (Thermo Fischer Scientific, Catalog # Hs00248508_m1), RGL2 (Thermo Fischer Scientific, Catalog # Hs01588058_g1), RGL3 (Thermo Fischer Scientific, Catalog # Hs01004328_g1), RALGAPA1 (Thermo Fischer Scientific, Hs04180554_g1), RALGAPA2 (Thermo Fischer Scientific, Hs00936528_m1), RALGAPB (Thermo Fischer Scientific, Hs00384265_m1), and TBP (Thermo Fischer Scientific, Catalog # Hs000427620_ma). RT-qPCR was performed using PerfeCTa FastMix II ROX (Quanta Biosciences, Beverly, MA, USA, Catalog # 95119-05K) according to the manufacturer’s instructions for Taqman assays. The cycling parameters were 50 °C for 2 min, then 95 °C for 10 min, followed by 40 cycles at 95 °C for 15 s and 60 °C for 1 min. Relative quantification of each gene was calculated and normalized to TBP expression.

### 2.9. Immunostaining and Quantification

Immunohistochemical (IHC) staining was performed on MDA-MB-468 tumors for platelet endothelial cell adhesion molecule [CD31/PECAM-1 (Cell signaling #77699, 1:3000)], Ki-67 [α-Ki67 (Abcam ab16667, 1:100)] and cleaved caspase-3 [CC3 (Cell Signaling Technologies, Danvers, MA, USA, 9661, 1:400)]. Slides were baked at 65 °C before deparaffinization in xylenes and rehydration. Antigen retrieval was performed using EDTA decloaker (Biocare Medical, LLC, Pacheco, CA, USA) in a pressure cooker at 95 °C for 2 min. Hydrogen peroxide (3%) was used to quench endogenous peroxidase before the tissues were blocked with DAKO protein block (Agilent Technologies Inc., Santa Clara, CA, USA). Sections were incubated overnight in the primary antibody at 4 °C. The following day, a biotinylated secondary antibody was added, and a Vectastain^®^ ABC (HRP) Kit and DAB substrate (Vector Laboratories, Inc., Burlingame, CA, USA) were used to develop the signal. Tissues were then dehydrated, counterstained with hematoxylin, cleared in xylenes, and mounted. IHC stained tumors were imaged on an PerkinElmer Vectra^®^ Automatic Quantitative Pathology Imaging System. Three representative photos were taken for each tumor at either an interior portion of the tumor or the stroma bordering the tumor. Images were then analyzed using inForm^®^ Advanced Image Analysis software version 2.2.1. The images were spectrally un-mixed, and the DAB signal was scored for the percentage of cells positive for CD31, CC3, or Ki-67 based on a user-defined threshold for positivity and normalized by inForm^®^ using the total cell number. The percent of cells within each scoring category was determined based on cell segmentation with the hematoxylin counterstain.

Immunofluorescence (IF) staining was performed on MDA-MB-468 tumors for Ki-67 [α-Ki67 (Abcam ab16667, 1:200)] and nucleoli (Novus, St. Louis, MO, USA, NBP2-32886, 1:250). Slides were baked at 65 °C before deparaffinization in xylenes and rehydration. Antigen retrieval was performed using EDTA decloaker (Biocare Medical, LLC, Pacheco, CA, USA) in a pressure cooker at 95 °C for 2 min. Tissue was blocked using 10% normal donkey serum (Jackson ImmunoResearch Laboratories Inc., Bar Harbor, ME, USA, 017-000-121), 5% bovine serum albumin (Research Products International, Mt. Prospect, IL, USA, A30075) and 0.3% Triton-X-100 (Fisher Scientific, BP151-100) in 1x PBS. Tumors were then blocked using a mouse-on-mouse block (Vector Laboratories, Newark, CA, USA, MKB-2213) to mask the endogenous mouse Ig. Ki-67 and nucleoli antibodies were then applied and incubated overnight at 4 °C. The following day, the 594 anti-mouse Alexafluor secondary antibody (Invitrogen, Waltham, MA, USA, A32744) and 488 anti-mouse Alexafluor secondary antibody (Invitrogen, A21202) were then applied. After washing, the nuclei were counterstained with Hoescht (Life Technologies Corp., Carlsbad, CA, USA, H3570) and mounted. Images were taken on a Zeiss LSM 800 confocal microscope using the Zen software version 2.3 (Carl Zeiss Microscopy, White Plains, NY, USA). Five images for each tumor were taken, and cells were hand counted for dual stained, Ki-67 only, or nucleoli only stained cells before averaging the populations for each tumor.

### 2.10. Collagen Quantification

Masson’s trichrome staining was performed using a ready-to-use kit (Abcam, Cambridge, UK). Briefly, tissue sections were deparaffinized, rehydrated, and heated in Bouin’s Fluid at 55 °C for 60 min. Subsequently, slides were washed in water before the nuclei were stained with Weigert’s Hematoxylin. After rinsing the slides in tap water, Biebrich Scarlet/Acid Fuchsin solution was applied for 15 min. Slides were then rinsed in water before differentiation in phosphomolybdic/phosphotungstic acid for 15 min prior to aniline blue staining for 10 min. Finally, slides were rinsed in water and incubated in 5% acetic acid solution for 5 min before the tissues were dehydrated, cleared in xylenes, and mounted. Trichrome-stained tumor samples were imaged on the PerkinElmer’s Vectra^®^ Automatic Quantitative Pathology Imaging System as described above. Analysis was performed using ImageJ Fiji Software version 20230710-2317 [37]. Briefly, regions of interest were drawn over the images to separate tumors from tumor-associated stroma before Masson’s Trichrome color deconvolution was performed. The green channel with collagen staining was selected, a constant threshold was applied to all images, and the percentage area stained was measured by ImageJ to obtain the tumoral and stromal percentage collagen.

### 2.11. Secreted Protein Array

The 80–90% confluent plates of MDA-MB-468 shCTRL, shRALA, or shRALB were washed three times with PBS before serum-free DMEM was added. A period of 24 h later, the media were collected, vortexed, and stored in Eppendorf tubes at −80 °C. Once three replicates had been produced, the media were thawed, spun down to remove any cellular debris, and one milliliter from each replicate was combined and vortexed before application to the Human Angiogenesis Antibody Array (Abcam, ab193655). Briefly, membranes were activated with the provided blocking buffer before one milliliter of cultured media was added to the blots overnight. Samples were then washed according to the manufacturer’s protocol before the application of a biotinylated antibody cocktail to the blots overnight. Membranes were washed and HRP-Conjugated Streptavidin was added to the blots overnight. After washing, membranes were imaged using the kit’s detection buffer solution.

Analysis of dot blots was conducted in ImageJ [38] using the Protein Array Analyzer macro. Briefly, images were converted into 8-bit before creating a grid on the blot so that each individual dot could be measured simultaneously. Expression values were then saved to Excel sheets, and the dots were normalized to positive control dots on each blot using the formula:X(Ny) = X(y) × P1/P(y)
where P1 is the mean signal from positive controls on a reference array, P(y) is the mean signal from positive controls on array “y”, and X(y) is the mean signal for spot “X” on array “y”. Normalized values for protein targets were then compared among blots with different cultured media applied to them.

### 2.12. ELISAs

ELISAs for ANGPT1 (Human Angiopoietin-1 Quantikine ELISA Kit, R&D Systems, Minneapolis, MN, USA) and CXCL1-3 (RayBio^®^ Human GRO ELISA Kit, RayBiotech, Norcross, GA, USA) were performed using CM from the MDA-MB-468 shCTRL, shRALA, and shRALB cells as per the manufacturers’ protocols. Standard curves and sample concentrations were calculated using the four-parameter logistic curve analysis on myassays.com, accessed on 13 November 2023. Data are presented as the average of three biological replicates +/−SEM.

### 2.13. Tissue Microarray Analysis

HER2 tissue microarrays (TMA) were constructed from archived breast tumor tissue isolated from 199 patients treated at the OSU James Comprehensive Cancer Center upon informed consent following approval from the OSU Institutional Review Board (IRB). The TMAs are maintained by the Columbus Breast Cancer Tissue Bank. Use of the TMAs was approved under IRB protocol #2016C0025.

IHC on BC patient TMAs was carried out using the Bond RX autostainer (Leica Biosystems, Inc., Nussloch, Germany). Briefly, slides were baked at 65 °C for 15 min and the automated system performed dewaxing, rehydration, antigen retrieval, blocking, primary antibody incubation with α-RALA (1:2000, #ab126627, Abcam), post primary antibody incubation, detection (DAB), and counterstaining. Samples were then removed from the machine, dehydrated, and mounted. Quantification and scoring of RALA immunostaining on BC patient TMAs were performed using inForm^®^ Advanced Image Analysis software (PerkinElmer, Waltham, MA, USA). InForm^®^ software was used to spectrally un-mix images, and the DAB signal was scored based on a user-defined threshold into four categories (0+, 1+, 2+, and 3+). The percent of cells within each scoring category was determined based on cell segmentation with the hematoxylin counterstain. An H-Score was calculated using the following formula: [1 × (%cells 1+) + 2 × (%cells 2+) + 3 × (%cells 3+)].

### 2.14. Western Blots

Protein samples were extracted from cells at 70% confluence in complete media, the concentration determined using the Pierce™ BCA Protein Assay Kit (Thermo Fisher Scientific), and Western blots were performed following the protocol described in Sizemore et al. 2007 [39]. Antibodies were purchased from Cell Signaling Technology (Danvers, MA, USA, RALA: 4799, RALB: 90879, GAPDH: G8795, anti-rabbit IgG HRP-linked antibody: 7074S, anti-mouse IgG HRP-linked antibody: 7076S) or Sigma (β-actin: A1978). Membranes were incubated in the RALA primary antibody incubated for 1 h at RT or in the RALB primary antibody overnight at 4 °C, GAPDH primary antibody incubation was for 1 h at RT, and β-actin primary antibody incubation was for 1 h at RT. Protein bands were visualized with the Immobilon ECL reagent (Millipore) using an Amersham Imager 600 instrument. Quantitation of relative protein expression by band densitometry was conducted using ImageJ software [38]. Original Western blot Data (uncropped blots) for is shown in Appendix A.

### 2.15. Pull-Down Assay

GTP pull-down assays were performed using a RAL-Activation Kit (Cytoskeleton, Inc., Denver, CO, USA, CAT #BK040). MDA-MB-468 cells were grown in DMEM with serum to 50–60% confluency, then serum starved for 48 h. MDA-MB-468 cells were pretreated with 50 µM of BQU57, RBC8, OSURALi, or an equivalent volume of DMSO for 1 h before being stimulated with recombinant EGF (Fisher Scientific, CAT# PHG0311) at a final concentration of 100 ng/mL for 2 min. Cells were lysed following the manufacturer’s recommendations. Pull-down was performed with 400 µg of lysate at a final concentration of 2 mg/mL with 10 µg of RALBP1 beads and placed on a rotator for 1 h at 4 °C. After the pull-downs were performed, 4× Laemmli Buffer with beta-mercaptoethanol was added, and the samples were boiled for 5 min at 98 °C before proceeding to Western blot. GTP-RALA and GTP-RALB were detected with Cell Signaling Technology antibodies (RALA: 4799, RALB: 90879) at 1:5000 and 1:2000 overnight, respectively. Quantification of the inhibited RAL activation was performed using ImageJ, and the data were normalized to the DMSO + EGF stimulated controls. RAL activation in the BC panel was measured using the MilliPore Sigma RAL Activation Kit (MilliPore Sigma, Burlington, MA, USA, CAT #17-300). Cells were grown to 70–80% confluency and washed twice with 1X PBS. RAL Activation Buffer was added, and lysates were incubated in glutathione agarose for 10 min (ThermoFisher, Cat #16100). Pull-downs were performed with 500 µg of protein lysate at 0.5 mg/mL, incubated with 10 µg of RALBP1 beads for 1 h at 4 °C, then washed with RAL Activation Buffer three times. After the pull-downs were completed, 4× Laemmli Sample Buffer with beta-mercaptoethanol was added, and the samples were boiled for 5 min at 98 °C. GTP-RALA and GTP-RALB were detected with Cell Signaling Technology antibodies (RALA: 4799, RALB: 90879) at 1:5000 and 1:2000 overnight, respectively, with both kits. To stimulate BT474 cells with heregulin prior to determining RAL-activation, cells were grown to 50–60% confluency, washed once with PBS, then serum starved for 48 h. After 48 h, cells were stimulated with NRG (ThermoFisher Scientific CAT#100-03-10UG) to a final concentration of 20 ng/mL and incubated for 10 min at 37 °C and 5% CO_2_. Following incubation, cells were processed for RAL activation as described above and Western blot. RAS activation assays were performed using the RAS Activation Kit (Cytoskeleton, Denver, CO, USA, CAT#BK008). In brief, 500 μg of protein lysate (final protein concentration of 0.5 mg/mL) was incubated with 30 μg of GST-RAF beads for 1 h at 4 °C with rotation. After the pull-downs were performed, 4× Laemmli Buffer with beta-mercaptoethanol was added, and the samples were boiled for 5 min at 98 °C before proceeding to Western blot. GTP-bound PAN-RAS was detected using the anti-PAN-RAS antibody (Cytoskeleton, #AESA02) at 1:250 in 5% milk TBST overnight. Quantification of RAS activity was performed using ImageJ.

### 2.16. Lead Compound Identification

Both drug-like and lead-like molecules were obtained from ZINC15 [40] by applying filters for molecular weight, logP, number of rotatable bonds, number of hydrogen bond acceptors/donors, and polar surface area (PSA). The properties were calculated using Schrodinger’s QikProp (Schrödinger Release 2016-2: QikProp, Schrödinger, LLC, New York, NY, USA). Pan-assay interference compounds (PAINS) were also removed from this set [41]. The final virtual screening included approximately 500,000 molecules. These molecules were converted to 3D structures and prepared for docking using Schrodinger’s LigPrep (Schrödinger Release 2016-2: LigPrep, Schrödinger, LLC).

The RALA-GDP crystal structure (2BOV) was obtained from the Protein Data Bank (PDB) [42] and used in the virtual screening (Appendix A). Since a crystal structure of RALB-GDP was not available at the time of study, RALB-GDP homology models were generated by RosettaCM (Appendix A) [43] using the RALA-GDP crystal structure as the template. The four best homology models out of the 5000 models generated were selected for virtual screening. All of the protein structures were prepared for docking by optimizing their hydrogen bond networks using Molprobity [44].

Virtual screening was performed targeting the allosteric binding pocket of the RALA-GDP crystal structure and the homology models of RALB-GDP using Autodock4 version 4.2 [45]. Docking was also conducted using the known RAL inhibitors RBC8 and BQU57 for comparison. The 54 highest scoring molecules determined by the predicted RALA-GDP binding were selected for validation.

### 2.17. Surface Plasmon Resonance

Surface plasmon resonance assays (Reaction Biology) were used to assess the RALA binding affinities for the selected compounds. A Series S SA sensor chip (Cytiva; formerly GE Healthcare; Marlborough, MA, USA, catalog # BR100531) was warmed from 4 °C to room temperature for approximately 30 min before docking the chip into the Biacore 8K instrument (Cytiva; formerly GE Healthcare). Once docked, the system was primed through the addition of immobilization running buffer (20 mM phosphate buffer with 2.7 mM KCl, 137 mM NaCl, 0.05% Tween 20, and 5 mM MgCl_2_; PBS-p+ with 5 mM MgCl_2_). The RALA protein used for this assay was site specifically biotinylated on the N-terminus for immobilization to the streptavidin functionalized sensor chip. The chip was pre-conditioned with 1 M NaCl/50 mM NaOH. The protein was then injected over the chip on the active flow cell (FC2) until a level of ~8000–9000 RU was obtained. A protein concentration of 25 µg/mL was used with a 600 s contact time and 5 μL/min flow rate. The flow system excluding the chip surface was then washed with a 50:50 mixture of 2 M NaCl/100 mM NaOH. Both the active and reference flow cells (FC2 and FC1) were then blocked with 0.1 mg/mL biotin to minimize non-specific binding to the chip surface.

Low molecular weight (LMW) multi-cycle kinetics/affinity methods were generated in the Biacore 8 K Control Software version 3.0.12.15655 (Cytiva). A solvent correction step was included to account for variability in DMSO. For the analyte, a contact time of 80 s was used with a dissociation time of 120 s. The flow rate was set at 30 μL/min. The running buffer was 20 mM phosphate buffer with 2.7 mM KCl, 137 mM NaCl, 0.05% Tween 20, and 5 mM MgCl_2_ with 1% DMSO. For each sample, ten concentrations were measured. The concentrations were injected from lowest to highest using twofold serial dilutions. Each sample was tested at least in duplicate.

Data analysis was performed using Biacore Insight Evaluation Software version 3.0.12.15655 (Cytiva). Data were double referenced (solvent corrected and reference subtracted) as well as blank subtracted. A steady state affinity model was applied to the data to obtain a KD via the formula:Req = ((C ∗ R_max)/(KD + C)) + offset
with Req as the steady state binding response, C as the analyte concentration, KD as the equilibrium dissociation constant, Rmax as the binding capacity of the surface (RU), and offset as the response at zero analyte concentration.

### 2.18. Patient Datasets

KM Plotter (kmplot.com, accessed on 16 June 2023) [46] was used to generate Kaplan–Meier survival plots that were analyzed by log-rank to assess statistical significance with relapse-free survival as the measured outcome. The dataset used for these analyses contained all patients meeting the St. Gallen criteria for BC subtyping in the KM Plotter metacohort. Correlation between RALA and RALB expression and HER2+ therapy response was analyzed using ROC Plotter (rocplot.org, accessed on 16 June 2023) [47].

Human BC patient data from the METABRIC invasive breast carcinoma cohort and The Cancer Genome Atlas BRCA cohort were obtained from cBioPortal. Each cohort of patients was subset by the PAM50 molecular intrinsic subtype, and only patients classified as TNBC (basal and claudin-low in METABRIC, basal-like in TCGA BRCA) or HER2+ were included for gene expression comparisons. Data cleanup and subsetting were conducted in R 4.2.2 utilizing dplyr 1.0.10 and tibble 3.1.8. Statistical significance was determined via unpaired Student’s *t*-test with Welch’s correction for unequal variance when applicable.

### 2.19. Data Analysis

Statistical analyses were conducted in GraphPad Prism 9.4.1. Statistical comparison of mean values was conducted via the unpaired *t* test, and statistical significance was set at *p* ≤ 0.05. Grubb’s test was performed to identify outlier values as appropriate.

EC50 values were calculated by generating a four-parameter variable slope regression to viability data in GraphPad Prism 9.4.1, and the maximum effect values were calculated by dividing the lowest best-fit values by the highest best-fit values [48,49].

## 3. Results

### 3.1. RALA Expression Is Predictive of Worse Patient Outcomes in TNBC and HER2+ BC

To evaluate the general prognostic importance of RALA and RALB in BC subtypes associated with RAS activation, we generated Kaplan–Meier survival plots using data from the KM Plotter breast cancer metacohort [46]. We focused on the luminal B, HER2+, and basal subtypes as defined by the St. Gallen criteria due to their association with increased RAS expression and activity [50,51,52]. Elevated RALA expression was associated with worse recurrence-free survival across all BC subtypes (*p* = 7.1 × 10^−9^, Appendix A) and in each individual subtype (Figure 1A upper panels. Luminal B, *p* = 4.8 × 10^−4^; HER2+, *p* = 0.012; and basal, *p* = 0.022). Conversely, high RALB expression was associated with better outcomes across BC (*p* = 8.5 × 10^−5^, Appendix A) and was not prognostic of survival within the individual subtypes tested (Figure 1A lower panels. Luminal B, *p* = 0.95; HER2+ BC, *p* = 0.27; basal BC, *p* = 0.58). Previously, our group reported that RALA expression but not RALB expression was predictive of BC response to chemotherapy [30]. Using ROC plotter for BC (rcplot.org, accessed on 16 June 2023) [47], we examined the significance of RALA and RALB expression as predictors of response to HER2-targeted therapies. Elevated RALA was associated with poorer response to HER2-targeted therapies, while RALB expression was not predictive of response to HER2-targeted therapies (Figure 1B). We next immunostained the TMA of samples from patients with HER2+ BC and found that high RALA immunostaining was associated with worse overall survival (*p* = 0.0090) (Figure 1C and Appendix A). Together, these data indicate that RALA and RALB differ as prognosticators of outcome in BC subtypes associated with increased RAS activity. While elevated RALA correlates with worse survival and treatment response in luminal B, HER2+, and basal/TNBC, RALB expression lacks prognostic or predictive significance in these subtypes.

### 3.2. Knockdown of RALA or RALB in the TNBC Cell Line MDA-MB-468 Reduces Tumor Growth and Alters the Tumor Microenvironment

We have previously shown that CRISPR knockout of RALA impaired, while knockout of RALB increased the viability of the TNBC cell line MDA-MB-231 both in vitro and in vivo [30]. To expand upon these findings, we used shRNA to stably knockdown RALA and RALB in the TNBC cell line MDA-MB-468. In contrast to our previous results in MDA-MB-231 cells, depletion of either RALA or RALB (Figure 2A) in the MDA-MB-468 cells increased the viability in 2D culture (Figure 2B), and the loss of RALA increased the viability in 3D culture while the depletion of RALB had no impact (Figure 2C). Migration (Figure 2D) and invasion (Appendix A) experiments did not reveal a role for RALA or RALB in these processes in the MDA-MB-468 cells. Surprisingly, when tested in vivo, we found that the loss of either RAL paralog significantly decreased the tumor growth in NSG mice (Figure 2E and Appendix A). These in vivo data, along with previously published findings, support a critical role for RALA in TNBC tumor growth and progression as well as a critical role for RALB in some, but not all TNBC models [19,30]. These observations are also in agreement with the strong correlation observed between elevated RALA expression and poor outcome in basal-like/TNBC and the inconsistent association between RALB expression and survival in these patients (Figure 1) [30].

To better understand how the loss of either RAL paralog decreased MDA-MB-468 growth in vivo while conversely increasing cell viability in vitro, we first used IHC to probe the tumors for expression of the proliferation marker Ki-67 and surprisingly found no difference in tumor proliferation between the groups (Figure 2F). Dual staining with antibodies to Ki67 and a human-specific nucleolin antibody confirmed that there were no differences in proliferation of the tumor cells between these groups when the tumors were harvested. IHC staining of tumors for expression of the apoptosis marker cleaved caspase 3 (CC3) revealed a slight decrease in apoptosis in the shRALA and shRALB tumors relative to shCTRL (Figure 2G). However, this observation did not explain how RAL depletion decreased tumor growth in this model, as diminished apoptosis would be expected to increase tumor growth. We next looked for changes in the tumor microenvironment (TME), which might explain the decreased tumor growth in the shRALA and shRALB groups. IHC for CD31/PECAM, a vascular marker, showed that shRALB tumors had decreased vasculature, both in the stroma surrounding the tumor (Figure 2H) and within the tumor area (Appendix A), relative to shCTRL or shRALA tumors. Next, Masson’s Trichrome was used to determine the collagen area in the tumors. Depletion of either RALA or RALB significantly increased the tumor collagen area relative to shCTRL tumors (Figure 2I). Differences in collagen distribution were also noticed between MDA-MB-468 shCTRL and RAL-depleted tumors. In shCTRL tumors, collagen tends to be organized into tight clusters or fibers (see arrow heads in Figure 2I, upper panel), while in shRALA and shRALB tumors, collagen is more uniformly distributed (Figure 2I, middle and lower panels). To understand how the loss of RALs in the tumor cells could impact the TME and thereby contribute to decreased tumor growth, we compared the secreted proteomes of the control and RALA or RALB depleted MDA-MB-468 cells. A human antibody array of proteins associated with angiogenesis (Appendix A) and confirmatory ELISAs revealed the secretion of angiogenesis regulator angiopoietin 1 (ANGPT1, Figure 2J), and the chemokine C-X-C motif ligand family 1–3 (CXCL1-3/GRO, Figure 2K) was significantly lower in the shRALA and shRALB conditioned media. These data suggest that in the MDA-MB-468 model, decreased tumor growth upon RAL depletion is driven by changes in the TME, with loss of RALB impairing both tumor vascularization and collagen deposition, and the loss of RALA primarily impacting the collagen deposition. These TME changes may be regulated by the secreted factors ANGPT1 and CXCL1–3.

### 3.3. Loss of RALA or RALB Does Not Affect In Vitro or In Vivo Growth of the HER2+ Cell Line SKBR3

Because elevated RALA expression was associated with poor outcome and treatment response in HER2+ BC patients (Figure 1), we also explored the impact of RAL knockdown upon in vivo and in vitro growth of the HER2+ cell line SKBR3. Here, stable knockdown of either RALA or RALB (Figure 3A) did not impact the orthotopic tumor growth (Figure 3B). Likewise, the viability of SKBR3 cells grown in 2D culture was not significantly changed by the depletion of either RAL (Figure 3C). As with our results in the MDA-MB-468 model, loss of either RAL increased the viability of the SKBR3 cells’ growth in low adhesion conditions (Figure 3D). We observed no changes in migration (Figure 3E) or invasion (Appendix A) phenotypes following RAL loss. None of the data support an important role for RAL in HER2+ cancers, as suggested by the patient data. It remains to be tested whether RALA imparts a more aggressive phenotype in HER2+ BC through a mechanism not tested by our immunocompromised mouse models and in vitro assays.

### 3.4. RALA and RALB Loss Negatively Impacts the Viability of TNBC, but Not HER2+, Cell Lines

In order to compare the effects of RALA and RALB loss across a broader group of TNBC and HER2+ BC models, we next used RNAi to transiently knock down the RALs in the TNBC cell lines (MDA-MB-231, BT-549, and MDA-MB-468), HER2+ lines (SKBR3 and MDA-MB-453), and a HER2-overexpressing luminal B line (BT-474) [53,54].

In vitro viability of these cells following the siRNA-mediated knockdown of RALA, RALB, or both paralogs was measured by the MTT and GILA assays. To mitigate the influence of off-target effects, three unique siRNA sequences were used for each RAL paralog, and knockdown efficiency was determined by Western blot (Appendix A). Our results indicate that simultaneous silencing of both RALA and RALB had a significant negative impact on the 2D (Figure 4A) and 3D (Figure 4B) viability of the TNBC lines tested. Surprisingly, simultaneous knockdown of both RALs did not consistently or appreciably decrease the viability in the HER2+ lines (Figure 4A,B). In fact, RAL knockdown increased the viability of the SKBR3 cells (Figure 4A,B). Transient knockdown of RALA alone tended to decrease the viability of TNBC cells but not to the extent of dual RAL knockdown (Appendix A). In the HER2+ cell lines, knockdown of RALA alone did not consistently decrease the viability and resulted in increased viability in some cases (Appendix A). Transient knockdown of RALB alone also consistently decreased the viability of the TNBC lines comparable to the effect of simultaneous RALA and RALB knockdown, but again, had a minimal impact on the HER2+ cell line viability (Appendix A). Based on these data, it is apparent that TNBC lines are more dependent upon RAL expression than the HER2+ lines, and both RAL paralogs support TNBC viability. Using these data, we classified the BC lines in our panel by RAL dependency, with the MDA-MB-231, BT-549, and MDA-MB-468 lines categorized as RAL-dependent, while the BT-474, SKBR3, and MDA-MB-453 lines were assigned as RAL-independent.

To determine whether the differences in viability following RAL knockdown correlated with RAL expression and/or activity, we performed Western blot analysis for RALA and RALB in lysates harvested from these cell lines (Figure 5A and Appendix A). RALA expression was highest in the MD-MB-468 cells and was significantly higher in these cells than in the MDA-MB-231, BT549, or MDA-MB-453 cells (Appendix A). However, when analyzed as groups, RALA was not consistently elevated in the TNBC cell lines relative to the HER2+ lines (Appendix A). The expression of RALB was not significantly different between any of the cell lines tested nor was the expression of RALB different between the TNBC and HER2 groups (Figure 5A and Appendix A). The activity of RALA but not RALB was significantly elevated in the TNBC cells relative to the HER2+ cell lines when these were analyzed as groups (Appendix A). Neither RALA nor RALB activity was significantly different when the cell lines were compared pairwise (Figure 5B,C).

The RALs are downstream effectors of RAS. RAS is mutated in the MDA-MB-231 TNBC cell line, and RAS can be activated in response to either EGFR or HER2 signaling, which are highly expressed in the TNBC and HER2+ cell lines respectively (Appendix A). We next asked whether RAL activity in the BC cell lines correlated with RAS activity. We tested this idea through RAS activity assays and surprisingly, we found no significant correlation between the RAS and RAL activity in these cell lines (Appendix A). Furthermore, when the BT474 HER2+ cells were stimulated by heregulin to activate HER3-HER2 signaling, the RAL activity was not increased while the phosphorylation of ERK was increased, indicating that RAS may activate RAF rather than RAL in these cells (Appendix A).

Given that RAL activity is dependent upon RALGEFs and RALGAPs, we postulated that low RALGEF or elevated RALGAP expression might explain the reduced RALA activity seen in the HER2+ lines. We evaluated the expression of the RALGEFs and RALGAPs across the cell lines by RT-qPCR (Figure 5D). Relative to the expression in the MDA-MB-231 cells, the HER2+ BT-474 cell line showed a high expression of RALGDS (3.3-fold), RGL2 (4.6-fold), and RGL3 (65.8-fold), and likewise, RGL3 was expressed at higher levels in HER2+ SKBR3s (8.3-fold) and MDA-MB-453s (13.5-fold; Figure 5D). Overall, the HER2+ lines expressed elevated levels of all RALGEFs other than RGL1, which was elevated in the TNBC lines. When we examined the expression of the RALGEFs in the METABRIC and TCGA BC patient cohorts, we also found that most RALGEFs (RALGDS, RGL2, RGL3, RALGPS1, RALGPS2) were significantly more highly expressed in HER2+ patients relative to the TNBC patients, while RGL1 was significantly upregulated in TNBC (Appendix A). These data do not support the idea that a lack of RALGEF expression generates reduced RALA activity in HER2+ lines. Upon the examination of RALGAP subunit expression, we found that the mean RALGAPA1 and RALGAPB expression was greater in the HER2+ lines relative to the TNBC lines while RALGAPA2 was not significantly different across the subtypes (Figure 5D). When the expression of the RALGAPs was analyzed in the METABRIC and TCGA BC cohorts, we found that RALGAPA1, RALGAPA2, and RALGAPB were all expressed at significantly greater levels in HER2+ BC relative to the TNBC patients (Appendix A). Thus, increased expression of some RALGAP subunits may contribute to decreased RALA activity in HER2+ lines.

### 3.5. The SMIs BQU57 and RBC8 Are Non-Specifically Cytotoxic to TNBC and HER2+ BC Cell Lines

Given that the genetic knockdown of the RALs reduced the viability of the TNBC cell lines but not the HER2+ cell lines, we hypothesized that TNBC would be more sensitive to experimental RAL inhibitors than the HER2+ lines. Surprisingly, we found that the HER2+ lines were at least as sensitive to RBC8 (Figure 6A) and BQU57 (Figure 6B) as the TNBC lines. The EC50s for RBC8 treatment in the TNBC lines were 53.34 µM (MDA-MB-231), 43.14 µM (BT-549), and 60.30 µM (MDA-MB-468), while the HER2+ lines had lower EC50s of 28.70 µM (BT-474), 25.72 µM (SKBR3), and 27.71 µM (MDA-MB-453). For BQU57, the EC50s of the TNBC and HER2+ lines displayed no clear trend by subtype: 115.1 µM (MDA-MB-231), 115.6 µM (BT-549), 119.8 (MDA-MB-468), 139.5 µM (BT-474), 86.37 µM (SKBR3), and 633.3 µM (MDA-MB-453). The cell lines were universally more sensitive to RBC8 than BQU57, but both drugs induced 99–100% maximum inhibition of cell viability in each line within the tested range (Figure 6). Unexpectedly, the TNBC lines, which all demonstrated RAL dependency upon siRNA knockdown (Figure 4), had higher RBC8 EC50 values and thus higher resistance to this inhibitor relative to the RAL-independent luminal B and HER2+ lines (Figure 6A and Appendix A) and BQU57 (Figure 6A and Appendix A). The incongruity between the insensitivity of the HER2+ cell lines to RAL knockdown and their sensitivity to RBC8 and BQU57 suggests that these inhibitors reduce cell viability through targets other than the RALs.

### 3.6. The Novel SMI OSURALi Binds RAL and Is Preferentially Cytotoxic to TNBC Relative to HER2+ BC and Normal Cell Lines

The results of our analyses of RBC8 and BQU57 in the TNBC and HER2+ cell lines reinforces the need for compounds that kill cancer cells in a RAL-dependent manner. We next evaluated the small molecule OSURALi as a novel RAL SMI. OSURALi was first identified through an in-silico screen of the ZINC small molecule library against the allosteric binding pocket of the GDP-bound RALA crystal structure (Appendix A). The binding of OSURALi to recombinant RALA was validated by SPR (Figure 7A and Appendix A). OSURALi was found to bind RALA with a KD of 0.948 µM. We next assessed the ability of OSURALi, RBC8, and BQU57 to inhibit RALA and RALB activation following EGF stimulation in the MDA-MB-468 cells (Figure 7B and Appendix A). Each of the three inhibitors significantly decreased RALA activation, and OSURALi induced the greatest mean decrease in activity relative to the EGF-stimulated control, although there was no statistical difference between the inhibitors (Figure 7B). RALB activation was significantly inhibited by BQU57 and OSURALi, and again, OSURALi treatment yielded the greatest mean decrease in activity but was not significantly more effective than the other inhibitors (Figure 7B).

We next determined the viability of the TNBC and HER2+ cell lines after treatment with increasing concentrations of OSURALi (Figure 7C). Unlike RBC8 and BQU57, OSURALi efficacy was substantially greater in TNBC relative to the HER2+ cell lines. This was especially noticeable when the maximum effect on the growth inhibition of each cell line by OSURALi was compared. The RAL-dependent TNBC cell lines showed 94.2–100% maximum growth inhibition while the RAL-independent HER2+ lines were associated with a far lower maximum growth inhibition ranging from 34.7 to 67.0% (Figure 7C). While the efficacy of OSURALi was significantly higher in the RAL-dependent lines relative to the RAL-independent lines, the potency of the drug, defined by the EC50, was not. The EC50s of the TNBC lines were 47.08 µM (MDA-MB-231), 20.65 µM (BT-549), and 31.22 µM (MDA-MB-468), while the HER2+ lines had EC50s of 11.13 µM (BT-474), 13.15 µM (SKBR3), and 31.26 µM (MDA-MB-453), respectively. The lower EC50s for the HER2+ lines were a result of their shallow inhibitory responses to OSURALi treatment (Figure 7C). The differences in sensitivity to OSURALi between the RAL-dependent TNBC and RAL-independent HER2+ lines were also markedly obvious when comparing the variability across lines treated with 75 µM or 100 µM of OSURALi (Figure 7D). Compared to the TNBC cells, OSURALi had a lower efficacy in the normal human endothelial-like cell line HMEC-1 compared to the RAL-dependent lines, with a maximum response of 53.3% growth inhibition while the EC50 in this line was 18.13 µM (Figure 7E). Taken together, these data show that the novel SMI OSURALi binds with high affinity to its RAL targets to inhibit their activity and kill cells in a RAL-dependent manner.

## 4. Discussion

The broad heterogeneity of BC necessitates the development of treatment strategies tailored to each molecular subtype, with TNBC remaining the most difficult subtype to target. The category of TNBC itself is represented by a diverse array of subclassifications [55], but treatment paths often converge into similar courses of cytotoxic chemotherapy due to a lack of actionable avenues for targeted treatment [3]. With high rates of metastasis and recurrence [56], the search for tractable therapeutic options for TNBC is of immediate importance. The paramount role of the RAL paralogs in the tumor-driving RAS signaling cascade and their amenability to small molecule inhibition makes them promising candidates for targeted anti-cancer therapies. Here, we report that RALA and RALB are essential for the viability of TNBC but not the HER2+ BC cell lines. These data support the hypothesis that RAS signaling in the TNBC and HER2+ cell lines make use of different downstream effectors (Figure 8) and further define the BC patient population that may ultimately benefit from RAL-targeting therapies. Additionally, we present the novel RAL inhibitor, OSURALi, as a potent alternative to the commercially available RAL inhibitors RBC8 and BQU57. We demonstrate that OSURALi, unlike RBC8 or BQU57, is more cytotoxic to RAL-dependent TNBC lines relative to the RAL-independent HER2+ and normal cell lines. Based on these data, further investigation and development of OSURALi as an anti-cancer therapy are warranted.

Here, we found that RALA mRNA expression correlates with poorer clinical outcomes across BC and specifically within the HER2+ and TNBC/basal BC subsets in the KM Plotter breast cancer metacohort. These data reinforce our previous results from the METABRIC and TCGA patient cohorts [30]. Likewise, through the immunostaining analysis of tissue samples from HER2+ BC patients, we found that elevated RALA immunohistochemical staining was associated with worse overall survival in this subtype, matching our previous results in a TNBC cohort [30]. Conversely, elevated RALB expression was associated with increased survival in BC patients in both our analysis of the KM Plotter metacohort presented here and in our prior analysis of the METABRIC population [30]. We can conclude that high RALA expression consistently predicts worse outcomes in BC while RALB, at least at the mRNA level, is a poor prognostic biomarker. It is also important to consider that while our analysis of the HER2+ patient data implicated RALA as a driver of poor outcome in this disease, our in vivo and in vitro studies did not corroborate this finding. It is possible that RALA regulates outcome in HER2+ patients through cell-extrinsic effects such as the suppression of immune function, which would not be discovered in our immune-incompetent mouse model and in vitro studies.

Our comparison of the effects of stable RALA and RALB silencing by shRNA presented several intriguing results. While the stable loss of either RAL in the TNBC MDA-MB-468 cell line was associated with increased viability in vitro, the loss of either RAL significantly decreased tumor growth following orthotopic injection into immunocompromised mice. These incongruent findings inspired us to investigate why RAL depletion produced the opposite effects on MDA-MB-468 tumor cell viability in vitro vs. in vivo. We hypothesize that RAL silencing may alter interactions between MDA-MB-468 tumors and the tumor microenvironment. It has previously been reported that RALA and RALB regulate pro-metastatic extracellular vesicle secretion in the 4T1 mouse mammary tumor model, and contradictory effects of RALB silencing, with RALB loss decreasing tumor volume in vivo but increasing proliferation in vitro, were also observed [19]. When staining MDA-MB-468 tumors for the endothelial marker CD31, we found that RALB but not RALA loss decreased CD31 expression within both tumoral and stromal regions, suggesting that RALB may support angiogenesis in vivo and its silencing may impede tumor growth. Analysis of the proteins secreted by our MDA-MB-468 control and the RALA and RALB depleted cell lines demonstrated that bFGF and ANGPT1 secretion decreased in the shRALA and shRALB cells relative to shCTRL, indicating that angiogenesis signaling is disrupted when either RAL is lost. Additionally, a Masson’s Trichrome stain of the MDA-MB-468 tumors suggests that loss of either RALA or RALB alters the collagen deposition, which is a phenotype associated with changes in BC tumor metabolism [57]. Taken together with the decreased TIMP-2 secretion observed in the shRALA and shRALB MDA-MB-468 conditioned media, the disrupted capacity of the knockdown cells to remodel the ECM may also contribute to their slower rate of growth. Given that the RALs are known to impact gene expression, exocytosis as well as exosome production and secretion, further studies are required to determine precisely how the RALs contribute to an altered tumor microenvironment. Furthermore, considering our observed changes in cytokine secretion following RALA and RALB loss in the MDA-MB-468 model, future experiments utilizing immunocompetent mice are needed to examine the role of the RALs in regulating the host anti-tumor immune response.

In contrast to the substantial negative effects on tumor growth observed when either RAL was silenced in the MDA-MB-468 cells, stable RALA and RALB silencing did not impact the in vitro or in vivo growth of the HER2+ cell line SKBR3. This unexpected observation led us to test whether the HER2+ cell lines had diminished dependency upon the RALs relative to TNBC cell lines. We measured the 2D and 3D viability of three TNBC cell lines (MDA-MB-231, BT-549, and MDA-MB-468), a luminal B line (BT-474), and two HER2+ lines (SKBR3 and MDA-MB-453) following transfection with siRNAs targeting the RALs. To address the potential influence of off-target silencing [58], we utilized three distinct sets of siRNA sequences targeting RALA or RALB for parallel comparison and confirmed isoform-specific RAL knockdown by Western blot. Our finding that the loss of both RAL paralogs generally led to decreased viability in the TNBC cell lines but not the HER2+ lines suggests that the importance of RALA and RALB signaling is subtype-dependent within BC. Within the TNBC lines, it was also interesting that RALB loss produced a more significant negative impact on viability than RALA loss. This is contrary to our observations that RALA expression is more prognostic than the RALB expression of BC patient outcomes. This incongruity may reflect the importance of RAL activity in addition to gross expression in driving BC phenotypes. We observed that both RALA and RALB activity was increased in TNBC relative to the HER2+ BC cell lines. It was also intriguing that the simultaneous depletion of RALA and RALB had a universally negative impact on cell viability in the TNBC cell lines tested. These data suggest that RAL-targeting therapies that impair the activity of both RALs may be efficacious, even in models where stable RAL knockout has demonstrated antagonistic roles for these paralogs.

While this work found key differences in RAL activity and dependency between the TNBC and HER2+ BC cell lines, the source of this discrepancy was not immediately obvious. One possible explanation is that HER2 overexpression is accompanied by a RAS-effector switch, leading to the activation of downstream RAS mediators other than the RALGEFs (Figure 8). The pattern of dimerization of EGFR family proteins including EGFR, HER2, HER3, and HER4 exerts a profound influence on the downstream signaling cascade they activate. As a result, changes in the relative degree of EGFR family protein expression and activation influence the effector pathways available within a cell [59]. Previously, other groups have shown that HER2-amplified BC cell lines can shift from PI3K/AKT pathway activation to RAS/MAPK signaling when grown in 3D rather than 2D culture conditions [60], and the suppression of Raf/MEK/ERK signaling resulting from Notch activation is capable of causing an effector pathway switch to RAL-1 signaling downstream of EGFR and RAS in C. elegans vulval patterning [61]. As the HER2+ lines we studied are less sensitive to the interruption of RALA and RALB signaling, we hypothesize that their tumor phenotypes are instead driven by a distinct pathway that may exist downstream of RAS, such as RAF/MEK/ERK, or be wholly independent of RAS signaling, such as the PI3K/AKT/MTOR pathway downstream of HER2/HER3 heterodimerization. This interpretation is supported by our observation that RAL activation is elevated in TNBC relative to the HER2+ cell lines. Interestingly, all three subunits of the RAL-inactivating RALGAPS were highly expressed in the HER2+ cell lines and patient samples relative to the TNBC cell lines and samples. Elevated RALGAP activity may, at least in part, explain HER2+ BC’s reduced dependency upon the RALs. It was also notable that only one RALGEF, RGL1, was more highly expressed in the TNBC cell lines and patients relative to their HER2+ counterparts. Additional studies of RGL1 as a driver of RAL activity in TNBC are warranted.

Previously, we reported that stable CRISPR knockout of RALA decreased, while stable knockout of RALB increased, the growth of MDA-MB-231 cells [30]. Here, we found that transient knockdown of either RAL decreased MDA-MB-231 viability. We hypothesize that this divergence between the effects of transient and stable RAL loss may be due to the CRISPR lines developing long-term compensatory adaptations to RAL signaling disruption. Differences between the effects of CRISPR knockout and siRNA knockdown have previously been identified as compensatory phenotypes may emerge in systems in which long-term silencing has been achieved, and off-target effects may occur following either strategy [62,63]. Our observed differences between the MDA-MB-231 CRISPR results and siRNA results may also have occurred due to the necessarily incomplete loss of RAL transcripts in the siRNA model, permitting a small but sustained degree of RALA and RALB production. Compensatory mechanisms circumventing RALA and RALB signaling may also have developed in our stably transfected shRALA and shRALB MDA-MB-468 lines during the selection process. This may provide an additional explanation as to why the stable knockdown of both RALA and RALB increased the in vitro viability, while transient siRNA knockdown of either paralog generally decreased the viability in this line. These results highlight the importance of studying the effects of both transient and stable depletion of the RALs in BC models. Whether RAL-targeting therapies mimic the effects of stable or transient RAL knockdown will be an important consideration when designing pre-clinical studies.

In our previous study of the RALs in the MDA-MB-231 TNBC model, we found that changes in tumor growth upon RAL paralog knockout were readily explained by changes in tumor proliferation [30]. RALA knockout resulted in decreased Ki67 tumor staining while RALB knockout was accompanied by increased Ki67 staining. These in vivo changes were also mimicked in vitro where RALA loss resulted in decreased viability, migration, and invasion while the loss of RALB did the converse. Thus, in the MDA-MB-231 model, the RALs appear to primarily influence tumor growth through cell-intrinsic mechanisms. Here, we found that in the MDA-MB-468 TNBC model, loss of the RALs decreased tumor growth in vivo without corresponding decreases to the tumor cell viability, migration, or invasion in vitro. Furthermore, decreased tumor growth was not accompanied by decreased Ki67 staining in the tumor cells, at least in tumors harvested 70 days post-inoculation. Instead, in the MDA-MB-468 model, we observed that decreased tumor growth was accompanied by changes in the TME such as decreased CD31 staining and changes to the collagen matrix along with alterations of the tumor secretome. This led us to conclude that in the MDA-MB-468 model, loss of the RALs decreases tumor growth primarily due to cell-extrinsic mechanisms. We also observed decreased CD31 staining indicative of reduced angiogenesis in MDA-MB-231 RALA-KO tumors (unpublished data), suggesting that this may be a universally important function of one or both RAL paralogs across tumor models. We speculate that the activating RAS mutation, which is an oncogenic driver in MDA-MB-231 cells, may also contribute to the cell-intrinsic differences that emerged between the MDA-MB-231 and MDA-MB-468 cells with stable RAL depletion. As the RALs are direct downstream effectors of RAS signaling, MDA-MB-231 cell survival after RAL knockdown may require compensatory mechanisms such as epithelial-to-mesenchymal transition, which influence their proliferation and migration.

Finally, we found that the cytotoxicity of BC cell lines to the commercially available RAL inhibitors BQU57 and RBC8 had surprisingly little correlation to their dependency on the RALs for viability. One explanation for this observation is that the cytotoxicity of these compounds is largely the result of off-target effects. Other groups have previously described the limitations of BQU57 or RBC8 as RAL inhibitors, citing poor reproducibility due to suspected chemical instability [36] and substantial inhibitory effects through non-specific pathways in murine platelet cells [35]. Here, we report that OSURALi, a novel small molecule inhibitor, is capable of binding RALA with high affinity, and like BQU57 and RBC8, is an effective inhibitor of RALA and RALB activation. Unlike BQU57 and RBC8, OSURALi demonstrates increased cytotoxicity in TNBC cell lines that are dependent on the RALs for survival relative to RAL-independent HER2+ cell lines and a normal cell line. It should be noted, however, that OSURALi still exhibits cytotoxicity in cell lines that are not RAL-dependent, indicating that this compound also has off-target effects. In addition, the potency of OSURALi is not significantly higher than other currently available RAL inhibitors. Much additional testing of OSURALi is needed to understand its potential limitations as a drug-like molecule and how these can be improved upon to eventually yield a RAL inhibitor suitable for clinical use. Given the importance of the RALs in metastatic BC [30], RAL-targeting therapies may significantly improve the outcome for TNBC patients. In addition to their potential as single-agent treatments, RAL-targeting therapies also have potential as adjuvants to current treatment regimens as we found that RALA may provide resistance to both chemotherapy [30] and HER2 therapy (Figure 1B). For these reasons, we propose further investigation of the anti-cancer properties of OSURALi and its continued development to provide a new therapeutic paradigm for TNBC.

## 5. Conclusions

While RAS is not frequently mutated in BC, elevated RAS activity drives aggressive BC phenotypes, and targeting RAS and its effectors may improve the outcomes for some BC subtypes. The RAL small G proteins are downstream effectors of RAS implicated in BC progression and metastasis. Here, we found that RAL activity was increased in TNBCs, and that this subtype was significantly more dependent upon the RAL paralogs for viability than HER2+ BCs. In addition, we found that the RALs contribute to TNBC tumor growth by altering the tumor microenvironment. Finally, we reported on a novel small molecule inhibitor of the RALs, OSURALi, which demonstrated elevated cytotoxicity in RAL-dependent TNBC cell lines. These studies further define the RALs as targets to improve outcomes for TNBC patients and provide a new compound for the development of RAL-targeting therapies.

## 6. Patents

The Ohio State University has filed a patent application on the use of the small inhibitor OSURALi. Parties interested in accessing this invention may contact Dr. Sizemore or OSU’s Technology Commercialization Office at innovate@osu.edu, referencing T2023-160.

## Figures and Tables

**Figure 1 cancers-16-03043-f001:**
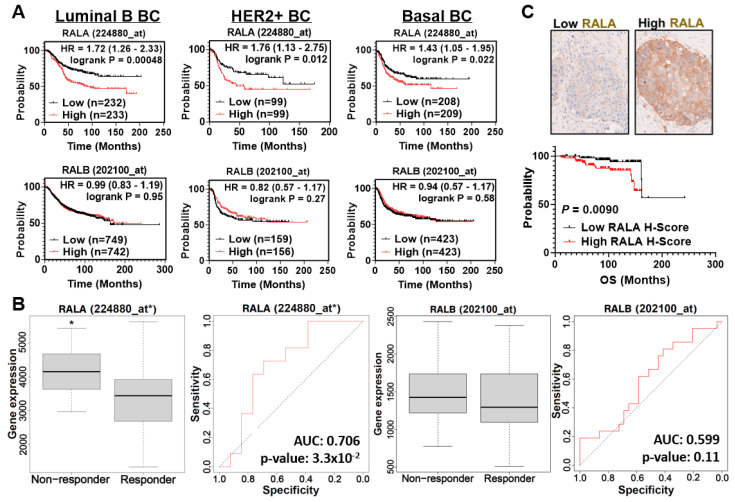
RALA is associated with poor outcomes in multiple breast cancer subtypes. (**A**) Kaplan–Meier analysis grouping BC patients within the luminal B, HER2+, and basal BC subtypes. Grouping was accomplished by segregation between high (upper 50th percentile) vs. low expression (lower 50th percentile) groups for RALA (upper panel) and RALB (lower panel) expression with relapse–free survival as the measured outcome. Data are from KM Plotter, *p* < 0.05. (**B**) RALA overexpression, but not RALB overexpression, predicts patient response to HER2 therapy. Five–year relapse free survival data were from ROC Plotter, *, *p* < 0.05. (**C**) Kaplan–Meier analysis of HER2+ BC patient overall survival (OS, *p* = 0.0090) as segregated by RALA H–score (upper 50th percentile vs. lower 50th percentile) based on patient sample immunostaining with representative images for samples with low and high RALA staining.

**Figure 2 cancers-16-03043-f002:**
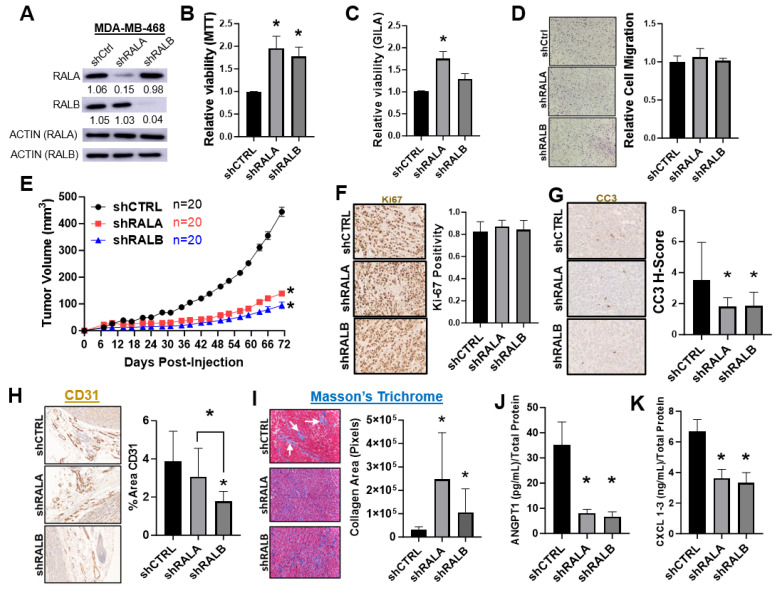
Depletion of RALA or RALB reduces MDA-MB-468 tumor growth and is associated with changes in the tumor microenvironment. (**A**) Western blots displaying RALA and RALB expression in the MDA-MB-468 shRNA control (shCTRL), shRALA, and shRALB cells. ImageJ was used for quantification. (**B**,**C**) Effects of stable knockdown of RALA or RALB in MDA-MB-468 cells on 2D growth as measured by MTT ((**B**), *n* = 4), and 3D growth as measured by GILA ((**C**), *n* = 4) *, *p* < 0.05. (**D**) Stable knockdown of RALA or RALB had no effect on MDA-MB-468 cell migration (left: representative images, right: composite mean values, *n* = 2). (**E**) Comparison of tumor growth following orthotopic mammary fat pad injection of MDA-MB-468 shCtrl (*n* = 20), shRALA (*n* = 20), or shRALB (*n* = 20). Results were combined from three independent experiments. (**F**–**H**) MDA-MB-468 shCTRL (*n* = 9), shRALA (*n* = 10), and shRALB (*n* = 10) tumors IHC stained for (**F**) Ki-67, (**G**) CC3, or (**H**) CD31. ROIs were determined on each image separating the tumor from stroma before color deconvolution to extract DAB staining. A signal threshold (equivalent for each image) was then applied to the samples before measurement of the ROIs was performed to measure the % area of target staining in each region. Three representative photos from each sample were separately analyzed, and the mean values were used for comparisons among groups. *, *p* < 0.05. (**I**) MDA-MB-468 tumors stained with Masson’s Trichrome, denoted by white arrows, to analyze the collagen deposition in shCTRL (*n* = 7), shRALA (*n* = 8), or shRALB (*n* = 9) MDA-MB-468 tumors. *, *p* < 0.05. (**J**,**K**) Graphs summarize the relative amounts of secreted proteins detected in the conditioned media from the MDA-MB-468 shCTRL, shRALA, and shRALB cultures.

**Figure 3 cancers-16-03043-f003:**
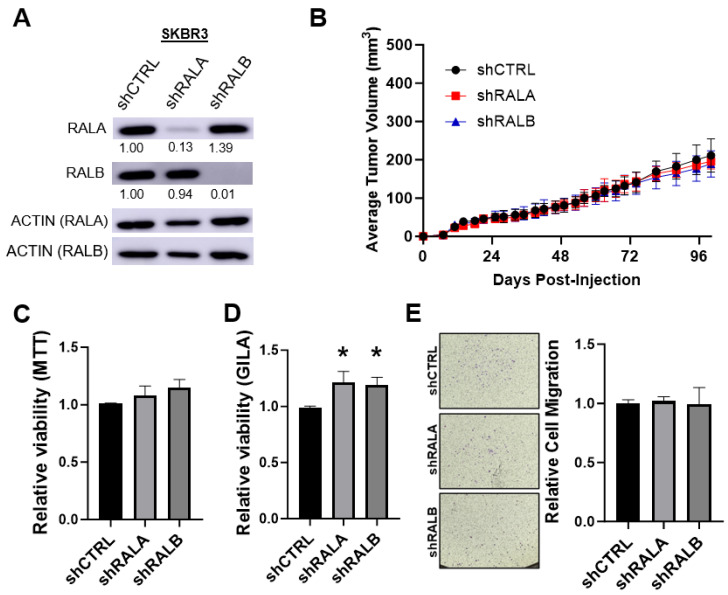
Loss of RALA or RALB does not negatively impact SKBR3 tumor growth or viability. (**A**) Western blots displaying RALA and RALB expression in the SKBR3 shRNA control (shCTRL), shRALA, and shRALB cells. ImageJ was used for quantification. (**B**) Comparison of tumor growth following orthotopic mammary fat pad injection of SKBR3 shCTRL (*n* = 10), shRALA (*n* = 9), or shRALB (*n* = 10). Results were from a single experiment. (**C**–**E**) Effects of stable knockdown of RALA or RALB in SKBR3 cells on 2D growth as measured by MTT ((**C**), *n* = 5), 3D growth as measured by GILA ((**D**), *n* = 5), and cell migration ((**E**), *n* = 2). *, *p* < 0.05.

**Figure 4 cancers-16-03043-f004:**
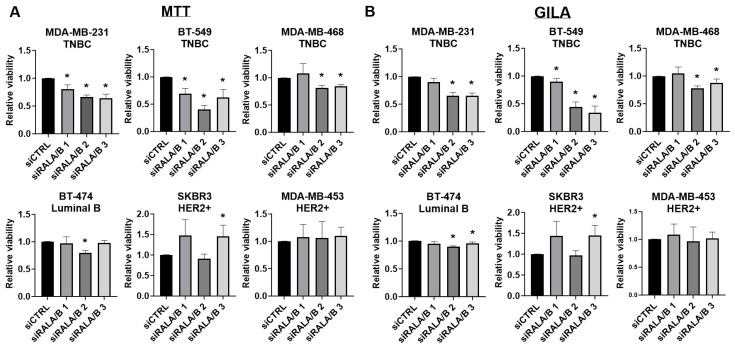
RAL depletion impairs the viability of TNBC but not HER2+ breast cancer cell lines. (**A**) Quantification of 2D growth over 3 days as measured by the MTT assay across a panel of BC cell lines grouped by molecular subtype. Cells were treated with the indicated pairs of siRNA sequences targeting RALA and RALB for 2 days prior to plating. *n* = 3–12. *, *p* < 0.05. (**B**) Quantification of 3D growth over 5 days as measured by GILA across a panel of BC cell lines grouped by molecular subtype. Cells were treated with the indicated pairs of siRNA sequences targeting RALA and RALB for 2 days prior to plating. *n* = 3–12. *, *p* < 0.05.

**Figure 5 cancers-16-03043-f005:**
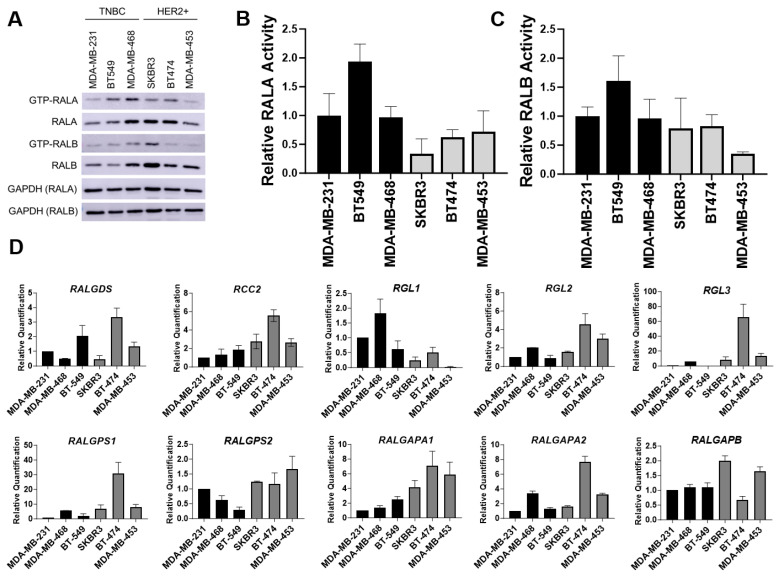
RAL activity is elevated in TNBC compared to HER2+ breast cancer cell lines. (**A**) Western blot of GTP-RALA, GTP-RALB, and total RALA and RALB expression in the TNBC and HER2 overexpressing cell lines. (**B**) Quantification of RALA activation in the TNBC and HER2 cell lines. Quantification was performed by averaging the normalized GTP-RALA/(t)RALA values in TNBC (MDA-MB-231, BT549, MDA-MB-468) or HER2 (SKBR3, BT474, MDA-MB-453) per experiment (*n* = 2). (**C**) Quantification of RALB expression by TNBC or HER2. Quantification was performed by averaging the normalized (t)RALB/GAPDH values in the TNBC (MDA-MB-231, BT549, MDA-MB-468) or HER2 (SKBR3, BT474, MDA-MB-453) lines from each experiment (*n* = 2). (**D**) Gene expression analysis was performed by qPCR to determine the levels of RAL GEFs (RALGDS, RCC2, RGL1, RGL2, RGL3, RALGPS1, and RALGPS2) and RAL GAPs (RALGAPA1, RALGAPA2, and RALGAPB) within a panel of BC cell lines normalized to MDA-MB-231 (*n* = 2–3). Four out of the seven RAL GEFs (RALGDS, RGL1, RGL2, and RGL3) were RAS activated RALGEFS. Expression above 1 indicates an increase in gene expression and expression below 1 indicates a decrease in gene expression compared to MDA-MB-231.

**Figure 6 cancers-16-03043-f006:**
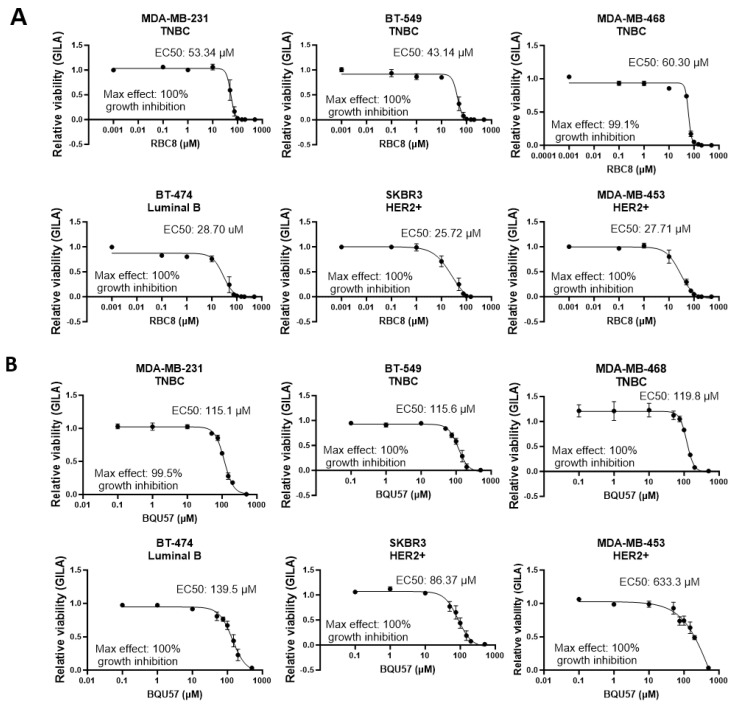
RBC8 and BQU57 treatment inhibits the in vitro viability in a panel of BC lines in a non-RAL dependent manner. (**A**) Relative viability of BC cells in 3D culture following treatment with the indicated concentrations of RBC8 for 5 days as measured by GILA (*n* = 3). (**B**) Relative viability of BC cells in 3D culture following treatment with the indicated concentrations of BQU57 for 5 days as measured by GILA (*n* = 3–4).

**Figure 7 cancers-16-03043-f007:**
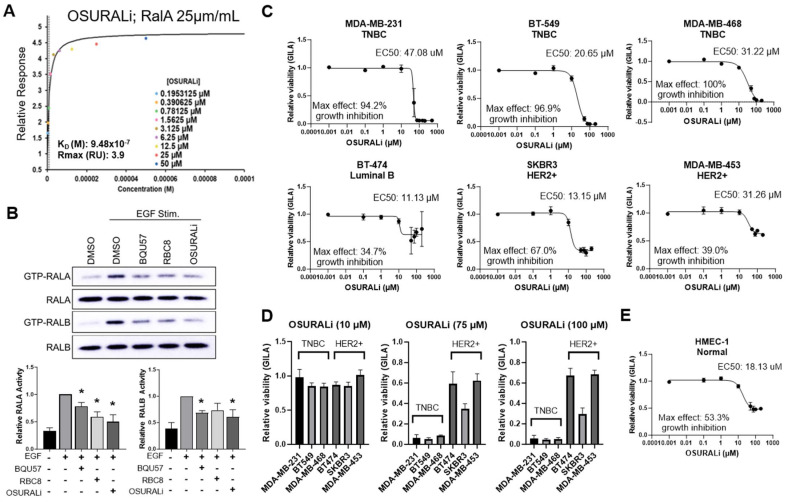
OSURALi inhibits RAL activation and is cytotoxic to RAL–dependent TNBC cell lines. (**A**) Response curve for OSURALi interaction with recombinant RALA. (**B**) Western blot (top) and quantitation (bottom, *n* = 4) of the inhibition of RALA and RALB activity by BQU57, RBC8, and OSURALi. MDA–MB–468 cells were pre–incubated with 50 μM of RAL inhibitor for 1 h prior to the stimulation of RAL activity by EGF. The novel RAL inhibitor OSURALi decreased the 3D culture viability to a greater degree in TNBC relative to the HER2+ BC lines. ImageJ was used for quantitation. *, *p* < 0.05. (**C**) Relative 3D viability of cells following treatment with varying concentrations of OSURALi for 5 days as measured by GILA (*n* = 3). (**D**) Viability data from Figure 7D displayed at selected concentrations to enable comparison by subtype. (**E**) Relative 3D (GILA) viability of HMEC–1 cells treated with increasing doses of OSURALi for 5 days (*n* = 2).

**Figure 8 cancers-16-03043-f008:**
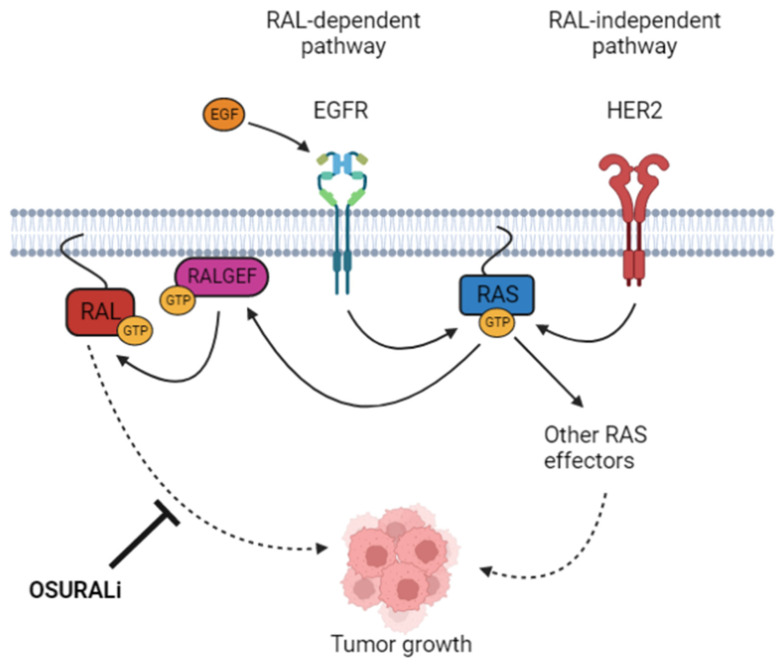
Proposed model of RAL-dependent and -independent signaling pathways through RAS. In TNBC, signaling through EGFR leads to the activation of RALs where they function to increase tumor growth. Conversely, in HER2+ BC, signaling through HER2 does not utilize RAL signaling and instead influences BC growth through the other RAS effectors. OSURALi will inhibit RAL activity, reducing TNBC, but not HER2+ BC proliferation. Figure was produced using Biorender.

## Data Availability

The original contributions presented in the study are included in the article/Appendix A, further inquiries can be directed to the corresponding author/s.

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
