# Peer review of "The RAL Small G Proteins Are Clinically Relevant Targets in Triple Negative Breast Cancer"

_cancers, 2024, doi:10.3390/cancers16173043_

Round 1

Reviewer 1 Report (Previous Reviewer 2)

Comments and Suggestions for Authors

The authors responded to the reviewers' comments at the previous stage of review and significantly revised the manuscript.

Small notes:

1. Figure 4 is better divided into 2 figures A and B, the inscriptions are too small and unreadable.

2. Figure 6 would be more clear if we combined the RAL-independent and dependent inhibition effects in one figure for each cell line.

Author Response

The authors responded to the reviewers' comments at the previous stage of review and significantly revised the manuscript.

Small notes:

  1. Figure 4 is better divided into 2 figures A and B, the inscriptions are too small and unreadable.

We thank the reviewer for their suggestion and apologize for our failure to address this request in our previous revision. We have again modified the figure to make it more easily readable. If our current efforts do not suffice, we are happy to split this figure as needed to ensure the readers can easily view the data.

  1. Figure 6 would be more clear if we combined the RAL-independent and dependent inhibition effects in one figure for each cell line.

We appreciate the reviewer’s comment. We have modified the text regarding the viability studies in Figure 6 and especially Figure 7 which we hope will make interpretation of these results easier for the reader.

Reviewer 2 Report (New Reviewer)

Comments and Suggestions for Authors

Here, the authors investigate the roles of RALA and RALB, two small G proteins activated by RAS, in different subtypes of breast cancer, especially triple negative (TNBC) and HER2-positive (HER2+). The authors show that TNBC cell lines are dependent on RAL expression for viability and tumor growth, while HER2+ cell lines are not. They also find that RAL depletion affects the tumor microenvironment in TNBC.The authors test the efficacy of two commercially available RAL inhibitors, RBC8 and BQU57, and find that they do not correlate with RAL dependency, suggesting that they kill cancer cells through other mechanisms. The authors report the discovery of a new small molecule inhibitor, OSURALi, which binds strongly to RALA, inhibits both RALA and RALB activation, and is more toxic to RAL-dependent TNBC cells than RAL-independent HER2+ and normal cells.

The authors replied to most of the comments of the reviewers, but some of the replies were vague.

- The authors did not provide any evidence for their hypothesis that the tumor growth was regulated by changes in the tumor microenvironment, which were not reflected in vitro. Could they discuss this point in the discussion, by for example discussing known work on RALA/ B, TBK1 activation and immune activation, for example citing: https://www.nature.com/articles/ncb2847 

- The authors did not address the discrepancy between the Ki67 staining and the tumor volume in the shRALA and shRALB tumors, and only speculated that the proliferation was slower at earlier time points. Could this be discussed further, as RALA/ B are implicated in various cell death mechanisms, for example citing; https://www.ncbi.nlm.nih.gov/pmc/articles/PMC7596570/

- The authors did not show the RAS activity in the different breast cancer cell lines, and only claimed that there was no correlation with the RAL activity. This can be assessed easily using DEPMAP, for example by analyzing RALA crispr screen results and dependency. see: https://depmap.org/portal/gene/RALA?tab=dependency&dependency=RNAi_merged

and correlate RALA and RAS crispr screens: https://depmap.org/portal/interactive/?filter=slice%2Fcontext%2FBreast%2Flabel&regressionLine=true&associationTable=false&x=slice%2FChronos_Combined%2F26574%2Fentity_id&y=slice%2FChronos_Combined%2F13749%2Fentity_id&color=

This could also be discussed at the end of the article

Author Response

The authors replied to most of the comments of the reviewers, but some of the replies were vague.

- The authors did not provide any evidence for their hypothesis that the tumor growth was regulated by changes in the tumor microenvironment, which were not reflected in vitro. Could they discuss this point in the discussion, by for example discussing known work on RALA/ B, TBK1 activation and immune activation, for example citing: https://www.nature.com/articles/ncb2847  

Thank you for this suggestion. We have now included additional text in the discussion on page 22 to provide readers with additional information regarding how the RALs may interact with the TME. The discussion includes the example citations and others on this topic.

- The authors did not address the discrepancy between the Ki67 staining and the tumor volume in the shRALA and shRALB tumors, and only speculated that the proliferation was slower at earlier time points. Could this be discussed further, as RALA/ B are implicated in various cell death mechanisms, for example citing; https://www.ncbi.nlm.nih.gov/pmc/articles/PMC7596570/

Again, we thank the reviewer for this excellent suggestion and have updated the discussion on page 24 to include a new paragraph on the known mechanisms through which the RALs may regulate cell viability and how these mechanisms could relate to our findings. We have included the above provided example and additional information on this topic.

- The authors did not show the RAS activity in the different breast cancer cell lines, and only claimed that there was no correlation with the RAL activity. This can be assessed easily using DEPMAP, for example by analyzing RALA crispr screen results and dependency. see: https://depmap.org/portal/gene/RALA?tab=dependency&dependency=RNAi_merged

and correlate RALA and RAS crispr screens: https://depmap.org/portal/interactive/?filter=slice%2Fcontext%2FBreast%2Flabel&regressionLine=true&associationTable=false&x=slice%2FChronos_Combined%2F26574%2Fentity_id&y=slice%2FChronos_Combined%2F13749%2Fentity_id&color=

This could also be discussed at the end of the article

We directly analyzed RAS activity in several TNBC and HER2+ breast cancer cell lines by pull-down assay. Examples of the pulldowns are shown in the western blots in what is now Supplemental Figure 9 (Pan-GTP RAs and Pan Ras). Correlation between experimentally determined RAS activity and experimentally determined RALA or RALB is shown in Supplemental Figure 9 C and D respectively. We modified the text on page 16 to make it clearer that we directly tested RAS activity in the breast cancer cell lines used and we have asked the editors to include figure legends for the supplemental figures in this round of review.

However, we appreciate that we can only perform activity assays on a limited number of samples and analyses of publicly available data might provide additional insight. As suggested, we have correlated dependency of RAL and RAS in the breast cancer cell lines in DepMap (see new Supplemental Figure 8C and D). We have also correlated expression of the RALs and RAS family members using gene expression data from METABRIC (Supplemental Figure 8A and B).

This manuscript is a resubmission of an earlier submission. The following is a list of the peer review reports and author responses from that submission.

Round 1

Reviewer 1 Report

Comments and Suggestions for Authors

This manuscript is an extension of this group’s previous study reporting the relevance of RALA in TNBC using the MDA-MB-231 cell line. The new data in this study is that the authors used several breast cancer cell lines including the TNBC, EGFR++ MDA-MB-468 cell line and HER2+ and luminal breast cancer cell lines to demonstrate that RALA is critical for TNBC and not HER2-type breast cancer. In addition, a new RAL inhibitor OSURALi is characterized. However, some of the data interpretation need to be addressed prior to publication.

·      In Figure 2, Mixing in vitro and in vivo data in the order of presentation is confusing. It is suggested that the in vitro data in cell lines is shown first, followed by the in vivo data in mice.

·      In figure 2 B, RALB was clearly more effective than RALA at tumor growth inhibition, Were these differences statistically significant? It is puzzling why this decrease in tumor growth is not reflected in the in vitro analysis of viability or migration.

·       Was metastasis evaluated in this study?

·      In Fig. 2, the KI-67 staining of tumor tissue shows no difference while tumor growth is clearly affected in the knockdowns. This reduced tumor growth in the absence of cell division needs to be explained. RALB is considered to affect angiogenesis, however, a consequence of increased blood supply is expected to be tumor cell growth and division.  Also in their previous publication, RALA KD significantly reduced Ki67 staining in MDA-MB-231 TNBC tumors, which paralleled the reduction in tumor volume. This discrepancy needs to be explained.

·      Fig 2J shows secreted proteins from 468 conditioned media. How many times was this experiment conducted and Why are there no error bars or statistical significance?

·      For cell migration, where N=2, how were the error bars calculated?

·      Fig. 5 A-B, RALA expression is considered to be consistent among the different BC cell lines. However, only a representative western is shown with varying degrees of expression. The quantification in figure S7 shows that MB-231 and BT549 TNBC cells express less RALA but the activity profiles are different. This quantification (quantified as a function of total RALA or RALB) with statistical significance between cell lines deemed to be more or less than other cell lines, should be shown in Figure 5 with representative western blots to form any definitive conclusions. Comparison with a non-cancer mammary epithelial cell line (HMEC) would indicate if BC cells overexpress RALA.

·      In figure 5C, are the differences between TNBC and HER2+ cells statistically significant?

·      Does the Ras activity and expression in these cell lines correspond to the observed differences in RALA/RALB activities?

·      In Figure 7, the new inhibitor OSURALi still inhibits the RAL-independent BT-474 and SKBR3 cell lines with lower EC50, albeit not by 100%. Does this mean that these viability effects are off target effects because knockdown of RALA and RALB did not inhibit viability?

·      Figure 7B shows that all inhibitors used reduce RALA/B activation to the same extent in the MB-468 cell line. However, the text states that “OSURALi induced the greatest mean decrease in activity relative to EGF-stimulated control (Figure 7B).“ Is this decrease in activity among the different inhibitors statistically significant?

·      Do all inhibitors used, reduce RALA/B activation (GTP binding) in the other RAL-dependent (in addition to MDA-MB-468), and also in cell lines designated “RAL-independent?”

·      The IC50 of OSURALi for RALA/B activation inhibition is not shown but is inferred from Fig. 7C to be similar to the existing inhibitors and is ~50 microM. The viability inhibition is also shown at >50 microM concentrations; therefore, the activity profile of OSURALi is too high to be pharmacologically useful.

·      The EC50 in HMEC cells of ~18 microM is lower than the EC50 for the TNBC “RAL-dependent” cell lines and is more comparable to the “RAL independent” breast cancer cells. However, the statement “Compared to TNBC cells, the normal human endothelial-like cell line HMEC-1 was also more tolerant of OSURALi, with a maximum response of 53.3% growth inhibition (Fig 7E),” needs to be clarified because it is actually more sensitive to OSURALi than the TNBC cell lines, in terms of the EC50. It is true that OSURALi does not achieve 100% cell death in this cell line; however, would 100% death be achieved at longer incubation times in OSURALi, thus indicating potential toxicity?

·      The conclusion that RALA/B preferentially acts through EGFR and not HER2 needs to be re-examined. The MDA-MB-468 is an EGFR overexpressing cell line. Do the other TNBC cell lines used express EGFR?  Also, if Heregulin was used as a chemoattractant in the HER2 expressing cell lines, would RALA/B not be activated by GTP binding, as indicated by the model?

Reviewer 2 Report

Comments and Suggestions for Authors

1. Figure 1a must be taken out as a separate one, because with such a small scale it is impossible to read the inscriptions.

2. For figures 2, 7, you should also zoom in, too small.

3. It seemed to me that the introduction was too long, in my opinion it can be shortened.
